# The Effect of Mice Adaptation Process on the Pathogenicity of Influenza A/South Africa/3626/2013 (H1N1)pdm09 Model Strain

**DOI:** 10.3390/ijms242417386

**Published:** 2023-12-12

**Authors:** Mohammad Al Farroukh, Irina Kiseleva, Ekaterina Stepanova, Ekaterina Bazhenova, Elena Krutikova, Artem Tkachev, Anna Chistyakova, Andrey Rekstin, Ludmila Puchkova, Larisa Rudenko

**Affiliations:** 1Federal State Budgetary Scientific Institution “Institute of Experimental Medicine”, St. Petersburg 197022, Russia; fedorova.iem@gmail.com (E.S.); sonya.01.08@mail.ru (E.B.); krutikova.iem@mail.ru (E.K.); anna.k.chistiakova@gmail.com (A.C.); arekstin@yandex.ru (A.R.); puchkovalv@yandex.ru (L.P.); vaccine@mail.ru (L.R.); 2Institute of Biomedical Systems and Biotechnology, Peter the Great St. Petersburg Polytechnic University, St. Petersburg 194021, Russia; mr.artem.tkachev.2001@mail.ru

**Keywords:** influenza A virus, A/South Africa/3626/2013 (H1N1)pdm09, H1N1pdm09 model strain, mutations in polymerase complex and hemagglutinin, pathogenicity, toxicity, immunogenicity

## Abstract

Influenza virus strain A/South Africa/3626/2013 (H1N1)pdm09 (SA-WT) is a non-mouse-adapted model strain that has naturally high pathogenic properties in mice. It has been suggested that the high pathogenicity of this strain for mice could be due to the three strain-specific substitutions in the polymerase complex (Q687R in PB1, N102T in PB2, and E358E/K heterogeneity in PB2). To evaluate the role of these replacements, SA-WT was passaged five times in mouse lungs, and the genome of the mouse-adapted version of the SA-WT strain (SA-M5) was sequenced. SA-M5 lost E358E/K heterogeneity and retained E358, which is the prevalent amino acid at this position among H1N1pdm09 strains. In addition, in the hemagglutinin of SA-M5, two heterogeneous substitutions (G155G/E and S190S/R) were identified. Both viruses, SA-M5 and SA-WT, were compared for their toxicity, ability to replicate, pathogenicity, and immunogenicity in mice. In mice infected with SA-M5 or SA-WT strains, toxicity, virus titer in pulmonary homogenates, and mouse survival did not differ significantly. In contrast, an increase in the immunogenicity of SA-M5 compared to SA-WT was observed. This increase could be due to the substitutions G155G/E and S190S/R in the HA of SA-M5. The prospects for using SA-M5 in studying the immunogenicity mechanisms were also discussed.

## 1. Introduction

Infection with the influenza virus is one of the most common diseases in the world, as it might infect 20% of the world’s population each season, which results in a death toll of 650 thousand people each year [1]. The main problem is that the influenza virus genome mutates at a high rate because its polymerase complex lacks a proofreading mechanism. This characteristic leads to the constant emergence of strains that are resistant to previously effective drugs and are not recognizable by the immune system that has been exposed to previous vaccine strains [2,3]. Thus, a mutation in the M2 gene, leading to the S31N substitution in the M2 protein, results in resistance to adamantanes, which are no longer recommended for treatment, since in 2013, 45% of influenza A strains became resistant to them [4].

The influenza A virus genome contains eight negative-sense, single-stranded RNA gene segments (HA, NA, M, PB1, PB2, PA, NS, and NP). Any mutation in one of these segments might lead to a dramatic change in the virus characteristics. Studying these mutations provides significant information to understand viral genome function and the role of each mutation in viral pathogenicity [2,5]. From this point of view, influenza virus strain A/South Africa/3626/2013 (H1N1)pdm09 (SA-WT) is noteworthy since it is highly pathogenic for lab animals (mice, ferrets). In other words, this strain has the required characteristics to be considered a model strain to infect these lab animals without the need for the adaptation process [6]. It was previously illustrated that this strain, compared to other H1N1pdm09 influenza A virus strains, has three strain-specific substitutions in its polymerase complex (N102T in PB2, E358E/K heterogeneity in PB2, and Q687R in PB1) [7]. The viral polymerase complex is encoded by three gene segments (PB1, PB2, PA) [8]. Mutations in these genes have been shown to play a critical role in strain pathogenicity, as the activity of the complex affects virus adaptation to mammals [9]. In addition, mutations in PB2 are responsible for virus sensitivity to host temperature, which alters its ability to reproduce in lower respiratory tracks [10,11]. Therefore, the mutations described by us might help to explain why this strain is highly pathogenic to mice without previous adaptation. We proposed that passaging in mice will leave only the variant most capable of replicating in mouse lungs, becoming the predominant population, which will give us more information about the effect of these strain-specific substitutions on virus pathogenicity.

In this work, a mouse-adapted version of SA-WT was obtained after five passages in mice (SA-M5). Subsequently, the differences in toxicity, lethality, reproduction, and immunogenicity of SA-WT and SA-M5 were discussed.

## 2. Results

### 2.1. Sequencing Analysis of SA-M5

To detect the changes in the three strain-specific substitutions (N102T in PB2, E358E/K heterogeneity in PB2, and Q687R in PB1), sequencing of PB2 and PB1 gene segments in the SA-M5 isolates was performed. Sequencing results revealed that after five sequential passages in mice, the only change that occurred was the disappearance of the A/G heterogeneity in the 1072nd position in the PB2 gene, which is responsible for E358E/K heterogeneity in the PB2 protein, and the G-variant, coding for E358, was the only variant in this position in all SA-M5 virus tested isolates (Figure 1).

The results of full-genome sequencing of SA-M5 revealed that SA-M5’s genome has no additional substitutions in any of its genetic segments except in the HA gene. In the HA gene, two heterogeneous positions (547R and 651M result in G155G/E variants and S190S/R, respectively) were found (Figure 2).

### 2.2. Viral Replication in Lung Tissue

Both strains, SA-WT and SA-M5, efficiently replicated in the lower respiratory tract of the infected mice, with no significant difference between the two strains. The titration of lung homogenates showed high titers for SA-WT and SA-M5, where the average titers were 8.06 ± 0.60 log_10_ EID_50_/mL and 7.74 ± 0.40 log_10_ EID_50_/mL, respectively (Figure 3).

### 2.3. SA-WT and SA-M5 Toxicity in Mice

To compare the toxicity of SA-WT and SA-M5, mice were inoculated intranasally with a high dose of fresh undiluted SA-WT or SA-M5 to cause lethality from acute pulmonary edema. In mice, infection with SA-WT and SA-M5 resulted in 80% and 70% lethality, respectively, within the first six days after inoculation with the viruses. No significant difference was observed between the two strains (Figure 4).

### 2.4. SA-WT and SA-M5 Pathogenicity in Mice

The survival study showed that both strains were highly pathogenic for mice with no significant difference between them, as mice lethality at the end of the experiment after intranasal inoculation with 5.4 log_10_ EID_50_/mL was 70% and 60% for SA-WT and SA-M5, respectively. Intranasal inoculation with 4.4 log_10_ EID_50_/mL caused 50% lethality for both strains. Even intranasal inoculation with 3.4 log_10_ EID_50_/mL caused 10% lethality for both strains (Figure 5). In addition, the LD_50_ values calculated by the Reed and Muench method [12] for both strains were almost identical, as 1 LD_50_ for SA-WT and 1 LD_50_ for SA-M5 were 4.60 log_10_ EID_50_/mL and 4.65 log_10_ EID_50_/mL, respectively.

### 2.5. SA-WT and SA-M5 Immunogenicity

To detect the serum antibody levels against SA-WT and SA-M5, a two-way hemagglutination inhibition test (HAI) was performed. Groups of mice infected with specified doses of the viruses SA-WT or SA-M5, described above, were used. Serum samples were collected from the surviving animals on day 21 post-infection. The sera’s abilities to inhibit the hemagglutination reaction of the two antigens (SA-WT and SA-M5) were evaluated. There were no antigenic differences detected in the two-way test between the two viruses as the values of the HAI tests were identical with both antigens (SA-WT and SA-M5) (Appendix A). HAI titers of SA-M5 sera, after mice inoculation with a series of infection doses, were significantly higher than titers of SA-WT sera (Figure 6).

## 3. Discussion

The presented work is a continuation of our investigations devoted to the analysis of biological properties of the influenza virus strain A/South Africa/3626/2013 (H1N1)pdm09 (SA-WT), which is a convenient model for laboratory studies of the mechanisms of influenza infection but remains unstudied. The SA-WT strain has three specific substitutions in the polymerase complex (N102T in PB2, E358E/K heterogeneity in PB2, and Q687R in PB1) [7]. Q687R localized in the C-terminus of PB1 that forms a strong bond with PB2 through its N-terminus. Hence, Q687R in PB1 might affect polymerase activity [7,13]. N102T in PB2 might also affect polymerase stability as the amino acid in this position is part of the PB2 subdomain that supports the PB1 thumb domain [14]. For E358E/K heterogeneity in PB2, our results correspond with literature data that this position is highly conservative in influenza subtypes that can infect humans and animals [15]. The 358th amino acid position in PB2 is part of the cap-binding domain; therefore, a mutation in this position might alter polymerase activity, affecting the strain’s pathogenicity [14]. Thus, each of the three substitutions of the polymerase complex may be important for the formation of the high pathogenicity of this strain.

The adaptation of a new human-origin influenza virus strain to mice usually leads to natural selection and gradual replacement of the initial strain with a more aggressive strain [16,17]. Since E358E/K heterogeneity was observed in PB2 in SA-WT, which means that there are two different copies of SA-WT in the virus population, the study was designed to allow the more pathogenic version to be the predominant copy in the population. In addition, the adaptation process involves acquiring new mutations that could appear in any of the eight gene segments of the influenza A virus [16], which might affect the other two strain-specific mutations: N102T in PB2 and Q687R in PB1.

The current study aimed to understand whether these substitutions play a role in the virus pathogenicity of SA-WT, hence, the investigation of whether these substitutions will disappear during the adaptation process and, if so, how this will affect viral pathogenicity. SA-WT was passaged in mice five times. After that, the predominant population in SA-M5 retained N102T in PB2 and Q687R in PB1 and lost the heterogeneity E358E/K in PB2 as SA-M5 contains only glutamic acid in the 358th position (E358) (Figure 1). However, further investigations showed no changes in pathogenicity as SA-M5 has the same level of pathogenicity as SA-WT, where the toxicity study showed 70% and 80% lethality for SA-M5 and SA-WT, respectively, with no significant difference (Figure 4 and Figure 5). The same observation was observed in their ability to replicate in the lower respiratory tract with average titers 8.06 ± 0.60 log_10_ EID_50_/mL versus 7.74 ± 0.40 log_10_ EID_50_/mL for SA-WT versus SA-M5 (Figure 3). In addition, both strains have almost the same 50% lethal dose (LD_50_) 4.60 log_10_ EID_50_/mL for SA-WT and 4.65 log_10_ EID_50_/mL for SA-M5. These data support our previous finding that N102T and Q687R substitutions might be the molecular basis that explains viral pathogenicity in mice [7]. In contrast, E358E/K heterogeneity in PB2 does not play a crucial role in strain pathogenicity as its disappearance did not affect the pathogenic characteristics of the strain.

Regarding the HA protein, it contains a binding site for sialic acid residues on the cellular membrane of the host cell and plays a remarkable role in the fusion process in the cytoplasm. In addition, it has a significant role in immunogenicity because it is the most abundant glycoprotein on the influenza envelope [18]. One of the primary viral strategies to avoid host response is mutations in the antigenic sites in HA [19]. These mutations have always drawn attention because they are the main reason that vaccines for influenza are changed annually. Unexpectedly, passaging in mice led to the appearance of mutations in the hemagglutinin (HA) gene. Two heterogeneities (G155G/E and S190S/R) were also identified (Figure 2).

SA-WT and SA-M5 differed in their HAI titers, which indicates that the immune response to SA-M5 was significantly higher than that against SA-WT. This difference can be explained by the two heterogeneities (G155G/E and S190S/R) in the HA of SA-M5. At amino acid position 155 in the HAs of H1N1pdm09 viruses, the G residue is the common residue, while the E residue is rare in the natural population. This position is also a part of antigenic site A [20]. The 155E variant had increased replication ability in epithelial cells (Calu-3 culture). The increased growth capacity in epithelial cells could be the cause of the increased immunogenicity of this variant in mouse models. It is interesting that in the two-way HAI test, the titers with different antigens were equal, which was possibly caused by the heterogeneity of the virus population. The 190 position is a part of the receptor binding pocket of HA. In addition, the R variant is usually a result of egg adaptation. The mechanism of egg adaptation was illustrated early [21,22]. It was shown that the S190R variant of the A/Hunan/26/2016 (H1N1)pdm09 virus had increased pathogenicity in mice [21]. However, in our study, the presence of this component in the population of the virus did not increase its pathogenicity (Figure 5).

The SA-M5 variant of SA-WT is perhaps even more attractive as a model virus than SA-WT itself for studying various aspects of the pathogenesis of influenza infection when screening anti-influenza chemotherapy drugs due to its high immunogenicity.

## 4. Materials and Methods

### 4.1. Viruses

Two virus strains were used in this study: 1—egg-derived A/South Africa/3626/2013 (H1N1)pdm09 (SA-WT): influenza virus strain A/South Africa/3626/2013 (H1N1)pdm09 was obtained from The Francis Crick Institute (London, UK) ID# 2014701384; 2—A/South Africa/3626/2013 (H1N1)pdm09 E1E2/M5 (SA-M5): was obtained from SA-WT strain after 5 passages in mice (see below Section 4.3)

### 4.2. Mice

Female CBA mice, aged 8–12 weeks, were kept in polycarbonate cages with free access to food and water.

### 4.3. Virus Adaptation to Mice

Five mice were lightly anesthetized with ether and inoculated intranasally with 50 μL of PBS containing 4.8 log_10_ EID_50_ of influenza SA-WT, divided equally between the nostrils. On the third day after infection, the mice were sacrificed, and their lungs were harvested in sterile conditions, after which they were homogenized using a small bead mill Tissue Lyser LT (QIAGEN, Hilden, Germany) in 1.0 mL of PBS containing antibiotic-antimycotic (Invitrogen, Waltham, MA, UK). A pool from the homogenates was prepared to perform the second passage of mice. After that, the pool was diluted 10-fold, and each mouse was inoculated with 50 μL of the diluted mixture. This process was repeated until the homogenized lungs from the fifth passage were obtained.

### 4.4. Viral Replication in Lung Tissue

For each of the two strains, SA-WT and SA-M5, a group of 10 mice was inoculated with the virus, and on the third day after infection, mice were sacrificed, and their lungs were harvested and homogenized in PBS. Homogenates were titrated in 10–11-day-old chicken embryos supplied by the “Sinyavino” poultry farm (Kirovsk Area, Leningrad region, Russia). Eggs were incubated for 72 h at 32 °C. The log_10_ EID_50_/mL calculation was based on the Reed and Muench method [12].

### 4.5. Viral Genome Sequencing

Samples were homogenized using a T10 basic ULTRA-TURRAX disintegrator (IKA, Staufen, Germany), and the total RNA was extracted and purified from homogenates of mouse organs by Rizol reagent (Diaem, Moscow, Russia). Viral RNA from chorioallantoic fluid was extracted by the precipitation method using a Ribo-prep kit (AmpliSens, Moscow, Russia). RT-PCR was performed with a Biomaser RT-PCR kit (Biolabmix, Novosibirsk, Russia) with electrophoretic detection of products and purification from the gel with a Clean-Up column kit (Evrogen, Moscow, Russia). Viral genome sequencing was performed with Applied Biosystems 3130xl Genetic Analyzer according to the manufacturer’s recommendations using BigDye™ Terminator v3.1 Cycle Sequencing Kit (Thermo Fischer Scientific, Waltham, MA, USA). The primers used for RT-PCR and subsequent sequencing are listed in Appendix A [23,24,25].

### 4.6. Toxicity in Mice

For each of the two strains, SA-WT and SA-M5, a group of 10 mice was inoculated with a high dose of fresh, undiluted virus under anesthesia with ether to study the virus’s ability to cause acute pulmonary edema, which is also known as the viral toxic effect, after which they were observed daily during the first six days post-infection to detect lethality from acute pulmonary edema as previously described [6].

### 4.7. Pathogenicity in Mice

A series of 10-fold dilutions was prepared in PBS for the model strain SA-WT and the strain SA-M5 obtained after five passages in mice. Using 10 mice per group, mice were inoculated intranasally under ether anesthesia with 50 μL of the previously prepared series. For the first 14 days post-infection, mouse lethality was observed, and the LD_50_ for each virus was calculated by the Reed and Muench method [12] and expressed as the log_10_ EID_50_/mL needed to give 1 LD_50_.

### 4.8. Hemagglutination Inhibition Test

A standard hemagglutination inhibition test was performed against four hemagglutination units (4 HAU) for both strains using sera obtained from mice that survived until the 21st day of the experiment. On the 21st day after infection, the mice were sacrificed, and blood serum was collected from each mouse separately. Subsequently, the sera were treated with RDE (Denka, Tokyo, Japan) and incubated for 1 h at 56 °C to inactivate the nonspecific hemagglutination inhibitors in the serum, after which the sera were incubated with 4% erythrocyte-PBS solution overnight at 4 °C to avoid nonspecific hemagglutination. Before incubation with viruses, sera were incubated for 1 h at 56 °C again and left to cool at room temperature. For each serum sample, a series of 2-fold dilutions were prepared using PBS containing either SA-WT 4 HAU or SA-M5 4 HAU, and then 1% erythrocyte-PBS solution was added. Subsequently, the agglutination of erythrocytes was observed, and hemagglutination inhibition titers were calculated accordingly [26,27].

### 4.9. 3D Structure

The UCSF Chimera 1.15 program was used to build the 3D structure. The 3D samples were obtained from Protein Data Bank PDB.

### 4.10. Ethics Statement

All of the experiments on mice and chicken embryos were performed according to European Union legislation [28], and the study was approved by the Institutional Local Ethical Committee (IEM, St. Petersburg, Russia). After each experiment, the animals were humanely euthanized.

### 4.11. Statistics

The GraphPad Prism 8 program was used to perform statistical analysis. Mouse survival was tested by the Mantel–Cox and Gehan–Breslow–Wilcoxon tests. For viral replication and the HAI test, the values of each group were initially tested for their normal distribution by the Kolmogorov–Smirnov test. After that, a two-tailed unpaired t-test with Welch’s correction was performed for normally distributed values, while the Mann–Whitney test was performed for values with no normal distribution. A *p*-value < 0.05 was considered statistically significant.

## 5. Conclusions

The adapted version (SA-M5) that was obtained after five passages in mice of influenza A virus A/South Africa/3626/2013 (H1N1)pdm09 (SA-WT) lost one of the three strain-specific substitutions, namely, E358E/K heterogeneity in PB2, without any change in its pathogenicity. These findings support the conclusion that E358E/K in PB2 does not play an essential role in virus pathogenicity. Additionally, the increase in the virus’s immunogenicity, which was noticed in the adapted version, could be explained by the two heterogeneities (G155G/E and S190S/R) that appeared in the HA. In light of these findings, SA-M5 is a more favorable model strain than SA-WT, as its higher pathogenicity could be a preferred factor when screening anti-influenza chemotherapy drugs due to its high immunogenicity.

## Figures and Tables

**Figure 1 ijms-24-17386-f001:**
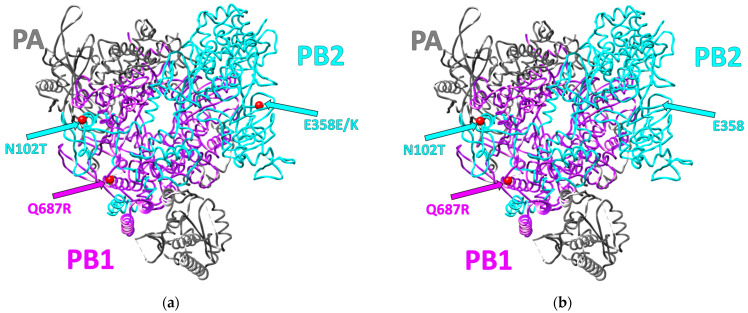
Three-dimensional structure of the polymerase complex of influenza A virus strains. (**a**) The polymerase complex of SA-WT compared to other influenza A(H1N1)pdm09 strains [7]. (**b**) The polymerase complex of SA-M5. PA subunits in gray, PB1 subunits in violet, and PB2 in light blue.

**Figure 2 ijms-24-17386-f002:**
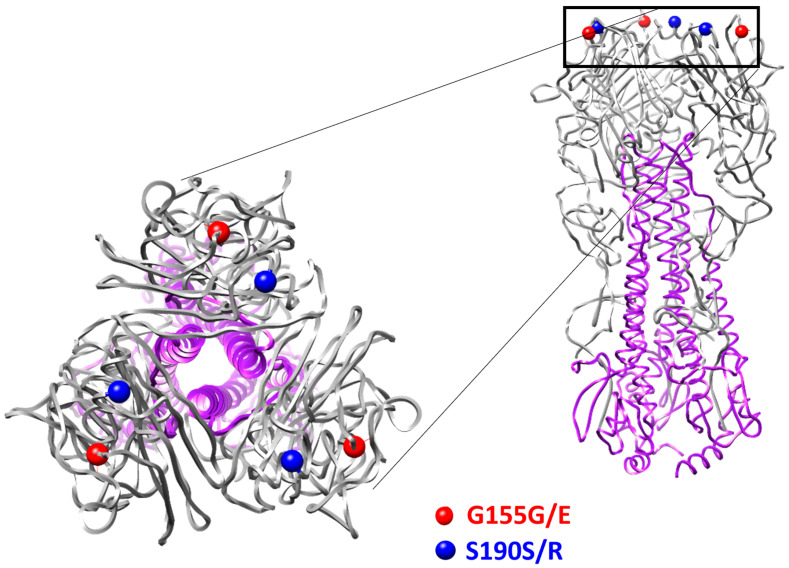
Three-dimensional structure of HA protein illustrating the two heterogeneities (155G/E and 190 S/R) that appeared after five sequential passages of SA-WT in mice (strain SA-M5). HA1 subunit in gray, HA2 subunit in violet.

**Figure 3 ijms-24-17386-f003:**
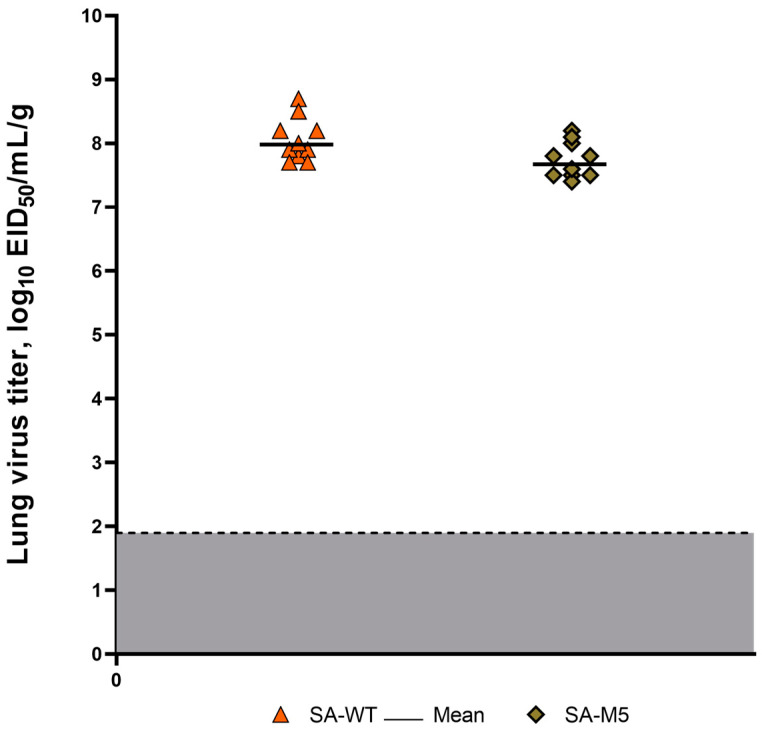
Virus replication in the lungs of mice harvested on the third day after the infection with SA-WT (orange triangles) and SA-M5 (green rhombus). The gray area presents the threshold of detection.

**Figure 4 ijms-24-17386-f004:**
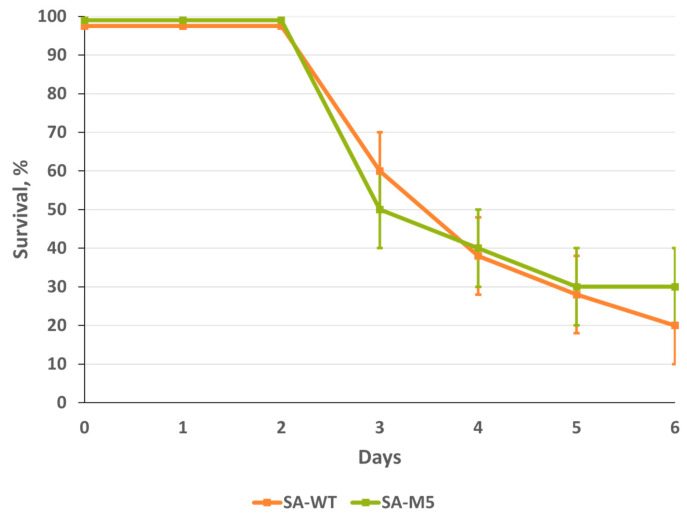
Mice lethality after six days of intranasal inoculation with toxic dose >7.5 log_10_ EID_50_/mL of SA-WT and SA-M5.

**Figure 5 ijms-24-17386-f005:**
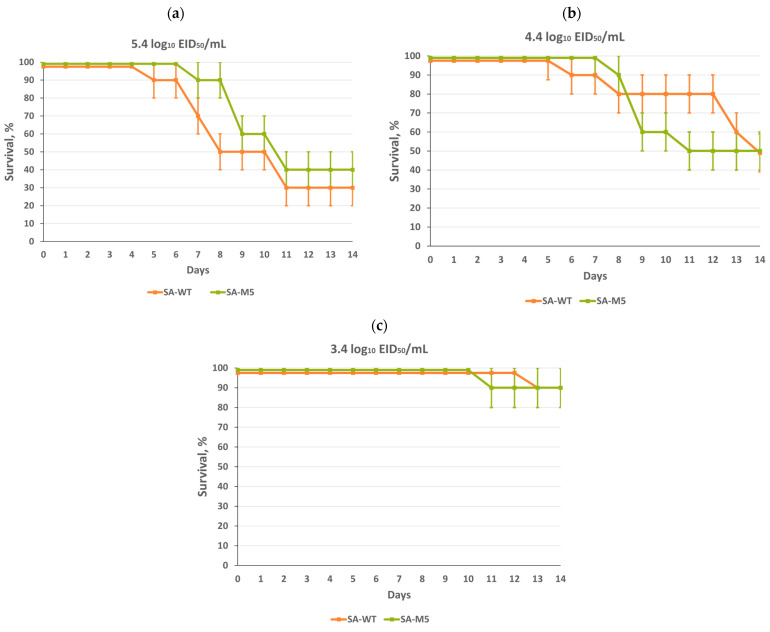
Survival in mice after intranasal inoculation with a 10-fold dilution series of SA-WT and SA-M5: survival in mice was monitored daily for 14 days. (**a**) Survival in mice after inoculation with 5.4 log_10_ EID_50_/mL, (**b**) survival in mice after inoculation with 4.4 log_10_ EID_50_/mL, and (**c**) survival in mice after inoculation with 3.4 log_10_ EID_50_/mL.

**Figure 6 ijms-24-17386-f006:**
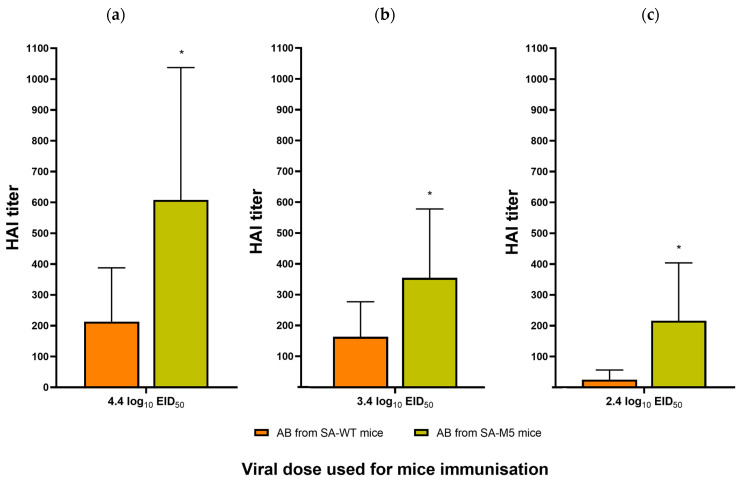
HAI assay of the tested sera incubated with SA-WT antigen. (**a**) Average HAI titers in blood serum after inoculation with 4.4 log_10_ EID_50_/mL were 608 ± 429 for SA-M5 sera and 213 ± 174 for SA-WT sera. (**b**) After inoculation with 3.4 log_10_ EID_50_/mL, the average titers were 355.5 ± 223 for SA-M5 sera and 164 ± 114 for SA-WT sera. (**c**) After inoculation with 2.4 log10 EID_50_/mL, the average titers were 216 ± 187 for SA-M5 sera and 25 ± 35 for SA-WT sera. Orange columns: mice inoculated with SA-WT. Green columns: mice inoculated with SA-M5. HAI values are presented as the mean ± SD. *—*p* < 0.05.

## Data Availability

Data are contained within the article and Appendix A.

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
