# Peer review of "The Effect of Mice Adaptation Process on the Pathogenicity of Influenza A/South Africa/3626/2013 (H1N1)pdm09 Model Strain"

_ijms, 2023, doi:10.3390/ijms242417386_

Round 1
Reviewer 1 Report
Comments and Suggestions for Authors
Farrouk and authors have demonstrated the differences in mouse passaged influenza virus. It is an interesting study, where authors have reported mutation in M5 variant. However, I have some concerns:
1. The last part of the results is very confusing. In supplementary figure, the values are exactly same for 2-way HAI assay. But in the main figure, the authors have shown significant differences. It is not clear what are the differences in methodology for these 2 set of experiments and how the assay was carried out.
2. In rest of the results section too, it will be better if authors can explain the results with some discussion rather than just putting the values.
Author Response
COMMENTS AND SUGGESTIONS FOR AUTHORS
Al Farroukh and authors have demonstrated the differences in mouse passaged influenza virus. It is an interesting study, where authors have reported mutation in M5 variant. However, I have some concerns:
Author’s response: Dear Reviewer #1, the authors sincerely thank you for your critiques and suggestions. Your valuable comments help us to improve the content of the manuscript. We greatly appreciate your positive assessment of our work. Please, find our answers to your comments below.
Point 1: The last part of the results is very confusing. In supplementary figure, the values are exactly same for 2–way HAI assay. But in the main figure, the authors have shown significant differences. It is not clear what are the differences in methodology for these 2 set of experiments and how the assay was carried out.
Author’s response 1: We greatly appreciate Reviewer #1 for this comment. We corrected the paper accordingly to clarify this issue. Figure 6 demonstrates the differences in the immunogenicity of two studied viruses, SA–WT and SA–M5. The HA antibody titers in SA–M5–infected mice were higher than in mice, infected with the same dose of the SA–WT virus (three doses were used 4.4, 3.4, and 2.4 log10 EID50/mL. To exclude the possible impact of antigenic differences we performed a standard two–way HAI test meaning the sera were tested with 4HAU of SA-WT (first antigen). They were also tested with 4HAU of SA-M5 (second antigen). The tests demonstrated that regardless of the antigen used for the HAI test the results were identical; therefore, we showed the result of only one HAI test (with 4HAU of SA-WT) Figure 6. Nevertheless, to make this point clearer we corrected the appropriate part of the Results section (3.5. SA-WT and SA-M5 immunogenicity, Figure 6, and its caption) (rows from 113 to 129). In addition, we modified Supplementary Table 2, and we added a description for it (rows from 4 to 12 in the Supplementary).
Point 2: In rest of the results section too, it will be better if authors can explain the results with some discussion rather than just putting the values.
Author’s response 2: The author thanks Reviewer #1 for the suggestion and criticism. However, this would turn the Results section into a Results and Discussion section. According to the instructions for authors of Int. J. Mol. Sci., the Discussion section should be presented separately from the Results section. If we had complied with your suggestion, we could have received a comment from the Editor about the incorrect formatting of the manuscript.
Reviewer 2 Report
Comments and Suggestions for Authors
The manuscript “The Effect of Mice Adaptation Process on the Pathogenicity of 2 Influenza A/South Africa/3626/2013 (H1N1)pdm09 Model 3 Strain” by Al Farroukh et al. characterizes the mouse adopted influenza SA-M5 virus strain in CBA mice.
The manuscript is well written, and the authors provided enough information about the need of their study.
Anyhow, there exist some major concerns about the study design per se as well as some major comments before this manuscript could be considered for acceptance.
The time interval of 3 d after infection for the 5 passages is way too short. This short time does not allow any selection pressure by the host immune system on the virus. A fact that is nicely demonstrated by your results – no effects...
Please provide an extra paragraph within the discussion section, about a better study design to select for either more or less aggressive mutants of the SA-WT virus strain.
To name the SA-M5 mutant “adopted” should by discussed – it should be avoided see above.
The hemagglutination inhibition assay needs N numbers. After 21 d of infection the serum was analyzed, but the previously data about the infected animals report only for 14 d.
Please provide a text for the respective table in the supplements.
Furthermore, this experiment is imprecise. Please determine serum levels of IgG, IgM, and monomeric IgA by ELISA. If it is necessary to focus on a single immunoglobulin for any reason – after 21 d I would assume the bigger effects still on IgM.
Why did you selected CBA mice? Please discuss.
Low N numbers of mice in all experiments reduce validity. Please discuss.
More information about the t-test is required. Was it a two-tailed students t-test?
Comments on the Quality of English Language-
Author Response
COMMENTS AND SUGGESTIONS FOR AUTHORS
The manuscript “The Effect of Mice Adaptation Process on the Pathogenicity of 2 Influenza A/South Africa/3626/2013 (H1N1)pdm09 Model 3 Strain” by Al Farroukh et al. characterizes the mouse adopted influenza SA–M5 virus strain in CBA mice. The manuscript is well written, and the authors provided enough information about the need of their study. Anyhow, there exist some major concerns about the study design per se as well as some major comments before this manuscript could be considered for acceptance.
Author’s response: Dear Reviewer #2, the authors sincerely thank you for all your critiques and suggestions. Your valuable comments help us to improve the content of the manuscript. We greatly appreciate your positive assessment of our work. Please, find our answers to your comments below.
Point 1: The time interval of 3 d after infection for the 5 passages is way too short. This short time does not allow any selection pressure by the host immune system on the virus. A fact that is nicely demonstrated by your results – no effects…
Author’s response 1: We thank Reviewer #2 for this comment. However, let us disagree with the respected Reviewer #2. Three days is the generally accepted standard time interval when, from the moment of infection, the titer of a live infectious virus reaches its maximum value. This is the basis of Pasteur's method of adaptation to another host. If the virus is allowed to remain in the body of an infected animal for a longer period, its infectious titer will progressively fall. As a result, at later stages of the experiment, we simply will not be able to isolate the live virus. Conversely, we did not need to allow the host immune system to exert any selective pressure on the virus. This selective pressure will kill the virus and prevent it from adapting to the host. We are interested in adaptive mutations that occur (or can occur) during each cycle of virus replication. The most optimal time for obtaining viral material for use in the next passage is 3–4 days post-infection. Please, also see our answer to point #3.
Point 2: Please provide an extra paragraph within the discussion section, about a better study design to select for either more or less aggressive mutants of the SA–WT virus strain.
Author’s response 2: We thank Reviewer #2 for criticism. Our main concern was about the heterogeneity of E358E/K in the gene encoding the PB2 polymerase subunit and its role in viral pathogenicity. We have established that it is not needed for the manifestation of the pathogenic properties of the virus, and the absence of a mutation in this position does not affect the properties of the SA–M5 virus in either direction. Nevertheless, a paragraph that justifies the classical virological approach used in the work is added to the Discussion (rows from 146 to 153).
Point 3: To name the SA–M5 mutant “adopted” should by discussed – it should be avoided see above.
Author’s response 3: we respectfully disagree with the opinion of the respected Reviewer #2. We believe that the term “mouse–adapted” is legitimate to use in such experiments as when describing the original egg–derived (egg–isolated) wild-type virus which has never "met mice" before, as "naturally mouse–adapted” and even more so for this virus, which has gone through a series of successive passages through the lungs of mice. The virus was ALREADY naturally adapted to mice, and the lethality rate among mice infected with this virus reached a record level, exceeding that of the well–known classical mouse–adapted virus PR8. We did not expect that within a few serial passages, the lethality of the passaged variant would increase even more. It was already very high. We were concerned about the heterogeneity of E358E/K in the gene encoding the PB2 polymerase subunit and its role in viral pathogenicity. We have established that it is not needed for the manifestation of the pathogenic properties of the virus, and the absence of a mutation in this position does not affect the properties of the SA–M5 virus in either direction. Please, also see our answer to the point #1.
Point 4: The hemagglutination inhibition assay needs N numbers. After 21 d of infection the serum was analyzed, but the previously data about the infected animals report only for 14 d.
Author’s response 4: We appreciate Reviewer #2 for this comment. Groups of 10 mice were infected with appropriate doses of two tested viruses and the lethality was monitored for 14 days. After this, the surviving animals were not sacrificed and blood was collected from surviving animals on day 21 post-infection. This information can be found in the sections 4.7 and 4.8.
Point 5: Please provide a text for the respective table in the supplements.
Author’s response 5: We appreciate Reviewer #2 for this suggestion. The following description was added to Supplementary Table 2: “Groups of 10 mice were infected with 3 different doses of two tested viruses and then lethality was monitored for 14 days. The surviving animals were used for serological study: the blood was collected from surviving animals on day 21 post-infection. In Table S2, the raw results of the two–way HAI test for each survived mouse are presented; the HAI values are indicated for each serum, for both antigens. There were no differences in antigenic properties detected for the two tested viruses: the HAI titers were the same for each serum regardless of the antigen used. Significant differences in the immunogenicity of the two tested viruses were detected; sera of animals infected with the SA–M5 virus had higher titers with both antigens (described in paragraph 3.5 of the Results section).” rows from 4 to 12 in the Supplementary.
Point 6: Furthermore, this experiment is imprecise. Please determine serum levels of IgG, IgM, and monomeric IgA by ELISA. If it is necessary to focus on a single immunoglobulin for any reason – after 21 d I would assume the bigger effects still on IgM.
Author’s response 6: We thank Reviewer #2 for this suggestion. However, in our study, we did not focus on the immunological status of mice after inoculation with SA–WT or SA–M5, so our tasks did not include the determination of all classes of immunoglobulins. The main task was to understand how the heterogeneity of E358E/K in the gene encoding the PB2 polymerase subunit could affect the pathogenicity of the virus. When using SA–WT as a model virus in experiments on mice, the mutation could disappear and it was necessary to understand, whether this would affect the level of pathogenicity of the virus or not; therefore, the consistency of the results. In addition, if after five passages in mice, the virus behaved in the HAI test in the same way as its original version, then, in this case, we would undertake more in–depth studies, including the enzyme–linked immunosorbent method ELISA. However, we immediately discovered a dramatic difference in the HAI titers of the virus before and after passaging in mice.
In theory, we could perform ELISA with sucrose–purified virus as an antigen, but we decided to choose the HAI test as a more representative method in our case. In ELISA with sucrose–purified virus, the role of anti–M and anti–NP antibodies is significant, but we aimed to detect differences in antibodies to hemagglutinin, which was detected to be heterogenic by sequencing. The recombinant HAs with appropriate differences are not available.
We are very grateful to Reviewer #2 for his valuable comment and in the future, as the next stage of work, we definitely will undertake detailed immunological studies that the respectful Reviewer #2 has recommended.
Point 7: Why did you selected CBA mice? Please discuss.
Author’s response 7: We appreciate Reviewer #2 for this comment. CBA mice, as well as some other lines (for example, BALB–c, white outbred mice, etc.), are an ideal model for studying the pathogenic properties of the influenza virus. Since the changes in mouse lethality from viral infection is one of the main quick and reliable criteria for assessing the viral pathogenicity, this approach was chosen in our study. In addition, this approach is widely used in primary screening of effective anti–influenza chemotherapy drugs and influenza vaccines and in several other studies.
Point 8: Low N numbers of mice in all experiments reduce validity. Please discuss.
Author’s response 8: We thank Reviewer #2 for criticism. Since during the process of virus adaptation to the animal’s body, the virus goes through dozens of cycles of replication, theoretically, even 2–3 mice could be enough for this kind of experiment (namely, for virus passaging in the body of mice). For each passage, we used 5 mice and believe that this number of animals is more than enough, especially since before carrying out a new passage, lung suspensions from mice were pooled. As for the study of the obtained passaged variant, of course, a certain minimum number of animals in the group is necessary. We strictly adhered to the 3Rs principles. Taking into consideration 3Rs concept, we used 10 mice in each group to evaluate the selected parameters (lethality, toxicity, infectious viral titer in the lungs, humoral immune response).
Point 9: More information about the t–test is required. Was it a two–tailed students t–test?
Author’s response 9: We thank Reviewer #2 for this comment. the corresponding tests (rows from 271 to 277) were added to the Manuscript. All statistical analyses were carried out using GraphPad Prizm 8. For survival studies, Log-rank (Mantel-Cox) and Gehan-Breslow-Wilcoxon tests were used. For viral replication and HAI test, the values of each group were initially tested for their normal distribution by Kolmogorov-Smirnov test. After that, a two-tailed unpaired t-test with Welch's correction was performed for normally distributed data while Mann-Whitney test was performed for data with no normal distribution.
Point 10: Comments on the quality of English language.
Author’s response 10: To improve the English language of the manuscript grammatical and punctuation errors have been fixed, the manuscript was proofread and edited by a virologist – native speaker, and Language Editing Services provided by Springer [https://authorservices.springernature.com/go/sn/?utm_source=Checklist_Writing+Quality&utm_medium=Website_Springer&utm_campaign=Platform+Experimentation+2022&utm_id=PE2022] and Grammarly [https://app.grammarly.com/].
Round 2
Reviewer 1 Report
Comments and Suggestions for Authors
Authors have answered all the comments and it has improved the quality of the work.